# A Quasi-Experimental Study of the Effects of Pre-Kindergarten Education on Pediatric Asthma

**DOI:** 10.3390/ijerph181910461

**Published:** 2021-10-05

**Authors:** Rie Masuda, Paul Lanier, Ellen Peisner-Feinberg, Hideki Hashimoto

**Affiliations:** 1Faculty of Political Science and Economics, Waseda University, Shinjuku, Tokyo 169-8050, Japan; r.masuda7@kurenai.waseda.jp; 2Department of Health and Social Behavior, School of Public Health, The University of Tokyo, Bunkyo, Tokyo 113-0033, Japan; hidehashim@m.u-tokyo.ac.jp; 3School of Social Work, The University of North Carolina at Chapel Hill, Chapel Hill, NC 27599, USA; 4School of Education, The University of North Carolina at Chapel Hill, Chapel Hill, NC 27599, USA; epeisner.feinberg@unc.edu

**Keywords:** Pre-K, child asthma, Medicaid, difference-in-differences, propensity score weighting

## Abstract

Ensuring access to pre-kindergarten (Pre-K) education remains a pressing policy issue in the United States. Prior research has shown the positive effects that Pre-K has on children’s cognitive development. However, studies on its effects on children’s health outcomes are scarce. This study aimed to investigate the effects of the Pre-K program on pediatric asthma. Children’s individual data from existing research conducted in North Carolina were linked with state Medicaid claims data from 2011–2017. There were 51,408 observations (person-month unit) of 279 children enrolled in Pre-K and 333 unenrolled children. Asthma was identified using the ICD 9/10 codes. A difference-in-differences model was adopted using a panel analysis with three time periods: before, during, and after Pre-K. The explanatory variables were interaction terms between Pre-K enrollment and (a) before vs. during period and (b) during vs. after period. The results indicated that children enrolled in Pre-K had a greater risk of asthma diagnosis during Pre-K (*b* = 0.0145, *p* = 0.058). Conversely, in the post-intervention period, the enrolled children had a lower of receiving an asthma diagnosis (*b* = −0.0216, *p* = 0.002). These findings indicate that Pre-K may increase the use of asthma-related health services in the short term and decrease the service use after participants leave the program.

## 1. Introduction

### 1.1. Background

Access to high-quality pre-primary education remains a longstanding and pressing policy issue in the United States and other countries [1]. Enhancing educational opportunities for preschool-age children has emerged as a policy priority because pre-primary education is an initial gateway to individual human capital building. Specifically, in the United States, federal funding for expanding pre-kindergarten (Pre-K) education for all 4-years-olds (i.e., universal Pre-K) has strong support in opinion polls [2] and is generally seen as a bipartisan political issue. Despite this broad support, enrollment rates and the quality of Pre-K programs differ across states and regions. Although the average nationwide enrollment rate of 4-year-olds in public-funded (i.e., state- or city-funded) Pre-K programs has more than doubled over the last two decades, in 2016 this rate was merely 32% in the United States [3].

Most prior evaluations of the effectiveness of Pre-K programs have focused on children’s school readiness, as represented by cognitive and non-cognitive skills. Although every public-funded Pre-K program is unique, most results have indicated that Pre-K has positive impacts on children’s cognitive skills (e.g., total test scores, language, and math skills) by the end of Pre-K or the early years in primary school, regardless of state [4,5,6,7,8,9,10]. In addition, some studies have reported public-funded Pre-K programs’ positive impacts on children’s non-cognitive skills, such as executive functions [8,9,10]. These encouraging findings regarding Pre-K’s effects on children’s cognitive and non-cognitive skills offer solid evidence supporting the expansion of opportunities for children to participate in Pre-K programs. 

Although ensuring children’s health is another important factor in human capital building, evidence concerning the short- and long-term impacts of pre-primary education programs on children’s health is scarce. For instance, two longitudinal follow-up studies demonstrated that attending intensive pre-primary education programs, which included routine home visits and parental education, were associated with a lower risk of lifestyle-related diseases and depression symptoms when the participants entered early adulthood [11,12,13]. It is important to evaluate the long-term and multidimensional health effects of pre-primary education because long-term health status affects all aspects of one’s life, such as educational attainment, adequate employment, and stable human relationships [14]. However, such studies of intensive, small-scale interventions conducted decades ago are not comparable to contemporary, large-scale, state-funded Pre-K programs. Studying Pre-K programs’ relatively *short-term* health impacts is also important because children’s good health status constitutes an important basis for their learning. However, few studies to date have empirically addressed the short-term effects of Pre-K programs on childhood health outcomes. A recent national heart study revealed that attending a universal Pre-K program in New York City was associated with increased diagnosis of several health problems (e.g., asthma, vision problems, and hearing problems) Similarly, an increased use of healthcare services (e.g., treatment for hearing or vision problems) was observed [15]. These findings suggest Pre-K programs may offer valuable settings for the early detection of children’s ill health status in children and promote the use of appropriate health care services. 

The present study focused on childhood asthma as an underdiagnosed, adverse health condition, for the following reasons. First, childhood asthma is an important public health problem because of its high prevalence, negative health consequences, and the disparity with which it occurs across different regions. In the United States in 2018, approximately 5.7% of children aged 0–5 years had ever been told that they had asthma or currently have asthma conditions [16]. In 2015, there were approximately 2.9 million asthma-related physician visits by children aged 0 to 17 years nationwide [17], representing an enormous amount of medical expenditure. Asthma also has short-term negative effects on children’s physical, social, and psychological functions [18]. It also has long-term negative effects on children’s general health status, obesity, missed school days, and missed work when they reach young adulthood [19]. The regional disparity of asthma attacks is also an important public health issue. The proportion of asthma attacks among children who have current asthma varies from 23% to 82% across states [20]. Second, the underdiagnosis of childhood asthma is a serious problem. Most childhood asthma symptoms develop before children turn 5 years old [21], yet 20–73% of children who currently have asthma symptoms remain undiagnosed [22]. Children with undiagnosed asthma tend to have lower quality-of-life scores and longer absences from school compared with children with diagnosed asthma [23]. Pre-K programs serve children who are at the precise age when asthma symptoms are first identifiable and proper medications can be initiated. Although asthma among preschool-aged children is often undiagnosed, when it is diagnosed it can be effectively managed with inhaled medication [24]. Thus, the potentially important role of Pre-K programs in detecting symptoms of asthma and referring children with these asthma symptoms to an appropriate health care service should be investigated. Third, previous studies of the relationship between daycare attendance and physician-diagnosed asthma have shown mixed results [25,26,27]. These contradictory results may be due to the cross-sectional or retrospective nature of the study designs that did not account for selection bias: whether a child enrolls in an early childcare program depends on their parents’ choice. Although these studies did not specifically address the association between Pre-K programs and physician-diagnosed asthma, the potential impact of pre-primary education program enrollment on subsequent childhood asthma should be more rigorously examined.

### 1.2. Underlying Mechanism

Some previous studies have sought to identify an underlying mechanism that explains how pre-primary education affects children’s health status [28,29,30]. Attending preschools may affect children’s health status through several pathways. First, attending pre-primary education increases the detection of diseases by teachers and licensed health professionals [31]. An earlier systematic review showed that mandatory health screening by pre-primary education may detect early symptoms of disease, which subsequently results in the early use of medical intervention [28]. Consequently, the early use of appropriate medical services may attenuate disease symptoms [32,33,34,35]. The detection effect that pre-primary education has is particularly crucial for detecting childhood asthma because asthma is often undiagnosed among preschool-aged children. Second, improvement in parent-related factors can affect every aspect of children’s lives, including their health status [31,32]. For example, parents who have children in school for longer periods are more likely to participate in the labor force and the better financial situation of households may improve the quality of the home environment. Parental respite enabled by children’s in-school time may also be associated with decreased parental stress related to child rearing. Third, the direct support of healthier meals and physical environments that pre-primary schools provide may affect children’s onset of asthma [29,30]. Tobacco-free air is a crucial factor in preventing childhood asthma [21]. For a child who is enrolled in Pre-K who would otherwise be exposed to smoke at home, the impact of clean air in pre-primary education settings could protect against asthma. 

### 1.3. Study Purpose

In the present study, we focused on the effects of one state’s Pre-K program on childhood asthma, which is one of the most common diseases among pre-primary-aged children. We differentiated the short-term effects of Pre-K programs on childhood asthma based on its immediate effects of the Pre-K program (i.e., detection effect) and the consequences after attending the Pre-K program (i.e., short-term health outcomes). Thus, we divided the study period into before, during, and after the period of attending the Pre-K program to better investigate the mechanism by which Pre-K program affects children’s relatively short-term health outcomes. 

We also differentiated asthma diagnoses into mild and severe asthma, as these may indicate different types of disease management. Specifically for childhood asthma, the severity of an asthma diagnosis can be an indicator of inappropriate disease management without continuous monitoring and the routine use of health care services [36]. Conversely, mild asthma may be an indicator that a child’s asthma is well controlled, even if the asthma symptoms are persistent [21]. Thus, severe asthma with exacerbations or asthmaticus should be differentially investigated apart from other mild asthma symptoms.

Given the potential importance of Pre-K for identifying and treating pediatric asthma conditions, the current study sought to test the following hypotheses: (a) Pre-K enrollment will increase asthma-related health service use due to a detection effect during Pre-K attendance, and (b) children who participate in Pre-K programs will have lower rates of asthma after they leave Pre-K programs because of early detection and subsequent routine medical care. 

## 2. Materials and Methods

### 2.1. Description of the Pre-K Program

The current study builds on an earlier evaluation of a statewide Pre-K program in North Carolina (NC). The evaluation study reported that children who attended NC Pre-K programs demonstrated better math skills and executive functioning (e.g., working memory) [8]. In NC, state-funded Pre-K programs meet all or most standard benchmarks set by the National Institute for Early Education Research each year [3]. This means that every Pre-K site in the state must satisfy these benchmarks, which included requirements for teacher-child ratio, class size, meal provision, and teacher accreditation. According to the program’s guidelines, the purpose of the program is to provide high-quality educational experiences that enhance children’s school readiness. The program provides 4-year-old children with classroom-based education for 6.5 hours per day, 180 days a year, as well as free breakfast, free lunch, and offers educational programs for the parents of these children. Specifically, NC Pre-K programs require that a health assessment conducted by licensed healthcare professionals be provided (by parents) within 30 days of a child’s enrollment and program directors are responsible for ensuring that any of their recommendations are followed [37]. The program specifically serves children from families with incomes below 75% of the state median income, dual-language speakers, and/or who need special education.

### 2.2. Data Source

We created a unique study dataset by linking data from the Pre-K evaluation study [8] with concurrent state Medicaid claims data. The evaluation study recruited 2843 families whose children attended kindergarten in the academic year 2015–2016 via a mailed survey using stratified random sampling of counties and school districts. Among the contacted families, 1524 families (54%) agreed to participate in the study, and 823 children were eligible for inclusion in the final evaluation sample. The inclusion criteria were: (a) not being retained in a school grade; (b) having a birthdate from September 1, 2009 to August 31, 2010; (c) not having an Individualized Education Program; (d) not attending a Pre-K site funded by the state’s Pre-K program (to avoid confounding treatment and comparison groups); and (e) having a family meeting the NC Pre-K income eligibility criteria. 

In the current study, the eligible sample data in the original evaluation study (*n* = 823) were linked to state Medicaid claims made from the beginning of 2011 to the end of 2017. Using names and dates of birth as keys, 612 children (74% of the evaluation sample) were linked to Medicaid claims data. Of the 612 children, 279 children were assigned to the treatment group, and 333 children were assigned to the control group. These data recorded health service use (indicated by the International Statistical Classification of Diseases and Related Health Problems (ICD)-9 and -10 codes), the type of service institutions used, and dates of service use. 

### 2.3. Measures

#### 2.3.1. Treatment

The treatment variable in this study is attendance in the state-funded Pre-K program in the academic year 2014–2015. The treatment group included children who had attended Pre-K as identified by the parent survey in 2015 and validated via the state administrative information for Pre-K attendance. The non-treatment group included children who had access to Pre-K but did not attend Pre-K.

#### 2.3.2. Outcome

Our primary outcome was all asthma diagnoses (i.e., codes beginning from “493” by ICD-9 and “J45” by ICD-10). Following a previous study that investigated a set of severe asthma diagnoses related to caregivers’ neglect or inappropriate disease management [36], eight diagnoses were identified by ICD-9 and -10 codes with exacerbation/status asthmaticus were categorized as severe asthma. These diagnoses were: (a) mild intermittent asthma with status asthmaticus; (b) mild intermittent asthma with (acute) exacerbation; (c) chronic obstructive pulmonary disease with acute lower respiratory infection; (d) chronic obstructive pulmonary disease with (acute) exacerbation; (e) exercise-induced bronchospasm; (f) cough variant asthma; (g) unspecified asthma with status asthmaticus; and (h) unspecified asthma with (acute) exacerbation. Other asthma diagnoses were categorized as mild asthma (see Appendix A).

#### 2.3.3. Period

We selected three time periods (i.e., before, during, and after Pre-K) to investigate the effect of Pre-K enrollment on asthma diagnosis and treatment. The before Pre-K exposure period was from January 2011 to August 2014; the during Pre-K exposure period was from September 2014 to August 2015; and the after Pre-K exposure period was from September 2015 to December 2017. 

#### 2.3.4. Family and Regional Characteristics

Twelve family characteristics were derived from the original Pre-K evaluation study: (a) children’s gender, (b) children’s race/ethnicity (white or non-white), (c) presence/absence of a chronic disease, (d) primary language spoken in the child’s home, (e) child’s first language, (f) parent’s first language, (g) the family’s military affiliation, (h) parental educational attainment, (i) family size, (j) number of adults in the household, (k) number of children in the household, and (l) family equivalent income. As a regional characteristic, population density by county (per square mile) in 2010 was calculated using U.S. census data in 2010 [38]. When a child moved from one county to another during the observation period (0.7% of the data), the population density of the county in which the child lived in at the beginning of the intervention period was used in the analysis.

To control for pre-existing medical conditions that might confound the potential association of Pre-K enrollment on children’s health, we categorized seven prevalent disease groups and health care service usage affecting children before the intervention period: neonatal issues, behavioral problems, metabolic diseases (e.g., diabetes), asthma-related conditions (i.e., obstructive pulmonary disease), abnormal weight (both overweight and underweight), and the frequency of routine health checkups (see Appendix B).

### 2.4. Analysis

Our study analyzed 84 months of data for each of the 612 children included in the study. The incidence of severe asthma was counted in units of person-months (i.e., a child diagnosed with severe asthma multiple times in a month counted as one person-month unit). Descriptive statistics on the demographic characteristics of children, households, and regions were gathered at the individual level. The incidence of asthma diagnoses was calculated by group (i.e., Pre-K group and non-Pre-K group) and by period (i.e., before, during, and after Pre-K enrollment).

As Pre-K attendance was self-selected, we conducted propensity score modeling to account for selection bias. We calculated propensity scores using a logit model that included 16 covariates that contributed to explaining the propensity of enrollment for NC Pre-K: (a) regional characteristics (i.e., population density in a child’s county of residence); (b) demographic characteristics of the family (i.e., child’s gender, a child having a chronic disease, child’s first language, family military affiliation, parental education, family size, number of children in the household, race/ethnicity, and family equivalent income); and (c) pre-existing diagnosis or use of health services (i.e., neonatal problems, metabolic diseases, obstructive pulmonary disease, overweight, underweight, and the frequency of health checkups). We selected these covariates because they have a demonstrable association with the likelihood of a child attending the Pre-K program. For example, children from families with a military affiliation were more likely to attend Pre-K, possibly because these families can use the service for free even if they have a higher income level than the eligibility criteria of NC Pre-K. To check the balance of the propensity of Pre-K enrollment of both groups, we calculated the standardized difference. After balancing, the two groups’ absolute standardized differences ranged from < 0.001 to 0.039, indicating fair comparability of the two groups (see Appendix C). We adopted the inverse probability weighting (IPW) in the following analysis.

For the main analysis, we conducted a panel analysis with a fixed-effect model using IPW and clustered by personal identifiers. Previous studies have recommended using propensity score weighting in the context of parametric Difference-in-Differences (DID) models to evaluate the causal effect of a policy or program [39]. Instead of dividing the study into two time periods—a practice employed in most DID analyses—following a previous study [40] we divided the study into three time periods: before the Pre-K period, during the Pre-K period, and after the Pre-K period. As we demonstrated Section 1.2, one of the plausible pathways showing how Pre-K attendance affects children’s health outcomes can be separated into two processes: (a) detection of disease and (b) subsequent treatment for the disease. These different processes at different time points that may affect children’s health outcomes could be differentiated by adopting a three-time-periods DID analysis that compares (a) before period vs. during period and (b) during period vs. after period.

Children’s ages (in months) at the time of their asthma diagnosis were included in the models to adjust for the varying prevalence of asthma at different ages. A 12-month dummy variable was also included to adjust for the varying seasonal prevalence of asthma. Finally, because family history and genetic heredity are recognized as strong predictors of asthma prevalence, a fixed-effect model was developed to account for time-invariant heterogeneity in the data.

The analytical formulation is as follows (1):*Y**_it_* = *β*1*Treated_it_* + *β*2*Before_it_* + *β*3*Treated_it_* × *Before_it_* + *β*4*Aftert_it_* + *β*5*Treated_it_* × *After_it_* + *β*6*Age_it_* + *β*7*Month_it_* + *u_it_* + *μ_i_*(1)

In this formulation, *Y* is asthma; *Treated* is children enrolled in Pre-K; *Before* is the period before Pre-K compared with the Pre-K period; *After* is the period after Pre-K compared with the Pre-K period; *Age* is the child’s age in months; *Month* is a dummy variable for each month when the diagnoses were made; *u* is the fixed effect, *μ* is the unobservable individual’s effect; *i* is an entity (i.e., an individual); and *t* is each time point. 

A few recent papers have voiced potential concerns that a propensity score matching/weighting method with DID analysis may generate a regression to the mean bias [41]. According to the findings, this bias should increase when there is a large pre-period difference in outcome level and weaker serial correlation in the outcome. As for pre-period differences in covariate levels, they do not generate biased estimations if they are not time-variant covariates and correlate with the outcome. To examine whether this regression to the mean bias occurs in our study, we conducted a DID analysis without using propensity score weighting (see Appendix E). To check the robustness of the main analysis, we conducted a few additional analyses. First, we analyzed hearing problems (i.e., starting from “38” in ICD 9 code) and vision problems (i.e., starting from “36” and “37” in ICD-9 codes) as outcomes using the same analytical model (see Appendix D). These outcomes may have a positive association with Pre-K attendance as shown in a previous study [15]. Second, we constructed three study subsamples based on the participants’ continuity of Medicaid enrollment because not all participants remained enrolled in Medicaid throughout the study period (i.e., from the beginning of 2011 to the end of 2017). The three groups included children who had been enrolled in Medicaid: (a) throughout the study period (i.e., from 2011 to 2017), (b) from the beginning of 2012 to the end of 2017, and (c) from the beginning of 2012 to the end of 2016. We then conducted analyses by subsample group using the same model as the main analysis (see Appendix F).

To determine the parallel trends assumption of our DID model, we conducted a parallel trend test for each outcome and found no significantly different trends between the treatment and non-treatment groups among the 44 observation points before the treatment period. We then plotted the time trend of each asthma diagnosis for both the treated and control groups during each observation period by trimester (see Appendix G, Appendix H and Appendix I). We identified no external shock (e.g., policy changes or economic recession) during the observation period. We conducted our analyses using the SAS statistical software package (version 8.4; SAS Institute Inc., Cary, NC, USA) and Stata on Unix (version 15; StataCorp., College Station, TX, USA).

## 3. Results

Table 1 presents the demographic and regional descriptions of the participants (*n* = 612) by group and the bivariate associations of all variables between groups. Child gender (male), family military affiliation, more children in the household, history of metabolic diseases, asthma-related conditions, higher frequency of child routine health examinations, and lower-population-density counties were associated with Pre-K enrollment.

Table 2 shows the proportion of asthma diagnoses in the person-month observation units over the three study periods. Before Pre-K, the treated group had a significantly higher risk for all asthma diagnoses than the non-treated group (2.03% vs. 1.34%, *p* < 0.001). During Pre-K, this significant gap increased (3.55% vs. 1.68%, *p* < 0.001). However, after Pre-K, the difference was no longer significant (1.83% vs. 2.17%, *p* = 0.119). For severe asthma, the proportions of diagnoses in the treatment and non-treatment groups were, respectively, 0.61% vs. 0.36% in the before period (*p* = 0.003), 1.02% vs. 0.40% in the during period (*p* = 0.001), and 0.87% vs. 0.74% in the after period (*p* = 0.340), respectively. For mild asthma, the proportion of diagnoses in the treatment group and non-treatment group was 1.60% vs. 1.15% in the before period (*p* = 0.001), 2.99% vs. 1.40% in during the period (*p* < 0.001), and 1.08% vs. 1.58% in the after period (*p* = 0.005) between the treatment group and the non-treatment group, respectively.

Table 3 provides the predicted coefficients of the interaction term of the treated groups and the treatment period from the panel analysis, adjusted for IPW. In Table 3, treated children before their Pre-K enrollment had a lower risk of all asthma diagnoses compared with non-treated children (Coef. = −0.0145, 95% CI (−0.0295, 0.0005)). This means that Pre-K children were more likely to be diagnosed with asthma during the Pre-K period than children not enrolled in Pre-K. After Pre-K, treated children were less likely to be diagnosed with asthma than non-treated children (Coef. = −0.0216, 95% CI (−0.0353, −0.0080)). Likewise, treated children before their Pre-K enrollment were less likely to be diagnosed with severe asthma compared with non-treated children (Coef. = −0.0064, 95% CI (−0.0116, −0.0006)), and treated children in the after period were less likely to be diagnosed with severe asthma than non-treated children (Coef. = −0.0061 95% CI (−0.0120, −0.0008)). Treated children before their Pre-K enrollment were less likely to be diagnosed with mild asthma when compared to non-treated children (Coef. = −0.0120, 95% CI (−0.026, 0.002)), and treated children in the after period were less likely to be diagnosed with mild asthma when compared to non-treated children (Coef. = −0.0185, 95% CI (−0.032, −0.005)).

When interpreting the coefficients of the analysis in Table 3, population attribution estimates can account for the health impact of Pre-K attendance. In 2015 in the U.S., there were 2,876,461 asthma-related physician visits made by children aged 0–17 years [17]. Given that one-eighteenth of the children were 4-year-olds, approximately 159,803 asthma-related physician visits were made by 4-year-old children in 2015. In addition, given that these visits occur in different months and were counted by person-month, we should expect 2317 person-month visits in a year (159,803 × 0.0145) should be referred to medical services during Pre-K attendance. Likewise, we should expect that Pre-K attendance would result in 3452 (159,803 × 0.0216) fewer person-month incidences in a year.

## 4. Discussion

### 4.1. Summarry of the Findings

The purpose of this study was to explore the effects of Pre-K participation on the rates of childhood asthma diagnoses in children. The findings indicate that Pre-K enrollment increased the likelihood that a child would be diagnosed with asthma during Pre-K and reduced the likelihood of an asthma diagnosis in the years following Pre-K enrollment. These findings suggest that Pre-K enrollment can help increase the early detection of asthma symptoms and that the subsequent use of asthma-related medical services can attenuate the prevalence of asthma after Pre-K. Children who enrolled in Pre-K had a higher likelihood of asthma diagnosis after they began attending Pre-K compared with children who never attended Pre-K. In contrast, Pre-K children had a lower risk for asthma diagnosis after they graduated Pre-K compared with their non-Pre-K counterparts. The magnitude of the decrease in asthma diagnoses after the Pre-K program was expected to exceed the increased detection cases during the Pre-K program.

Our findings demonstrated that the effects of Pre-K on pediatric asthma diagnosis can be discriminated by time period. The three-time-point DID model we adopted revealed that Pre-K enrollment increased rates of asthma diagnosis and decreased rates of asthma-related health service use among children who attended Pre-K. The increase in asthma diagnosis during Pre-K period may be attributable to the detection effect of Pre-K. Teachers’ observations and mandated health assessments by licensed health professionals might be crucial occasions for detecting pertinent symptoms such as coughing, wheezing, dyspnea, and chest tightness. Although early childhood schoolteachers are not health professionals, they often have training in child development and likely acquire some knowledge of prevalent diseases that could affect children in early education settings [37,42]. They can also communicate with caregivers directly and provide referrals to healthcare service providers who can provide caregivers with greater consultation of their child’s symptoms. Although the main purpose of Pre-K programs is to build a strong foundation for children’s school readiness, Pre-K teachers’ and health professionals’ abilities to detect diseases in children and connect their caregivers to relevant health services is another beneficial side effect of Pre-K programs. 

The effects on childhood asthma after children leave the Pre-K program possibly reflect (a) a decrease in the real incidence of severe asthma and (b) a reduced utilization of health services due to the reduction of symptoms. Children who enrolled in the Pre-K program received a detection effect 1 year earlier than their non-enrolled counterparts because (in principle) all children should be equally observed by school professionals when they begin kindergarten. If Pre-K enrollment allows the professionals’ detection of asthma symptoms to begin 1 year earlier, the increase in the diagnosis rate during Pre-K attendance should be offset when the non-Pre-K group enters kindergarten. If this is the case, the detection effect of Pre-K programs on severe asthma might be offset when non-Pre-K groups start attending kindergarten. 

However, the decrease in all asthma and mild asthma diagnosis rates after the Pre-K period among the children who attended the Pre-K program remained, withstanding the offset possibly caused by kindergarten’s detection effect. A likely explanation for this decrease may be that the effect of Pre-K on the real incidence of severe asthma might not be clear until years after children leave Pre-K because asthma is a chronic disease that requires long-term routine monitoring and medication [21]. Due to teachers’ and health professionals’ advice, caregivers of children enrolled in Pre-K possibly started routine monitoring of asthma symptoms earlier whereas caregivers of children who did not attend Pre-K might start such routine monitoring much later in their children’s lives [43]. Attending Pre-K can have preventive effects on asthma in children because of the early detection and early treatment that it facilitates. Earlier detection of asthma symptoms is also important for enhancing children’s well-being because many asthma symptoms are often overlooked, which results in children’s deteriorated quality of life [23]. However, in the current study, the observation period after the Pre-K period is limited to 2.5 years. Another study with a longer observation period is needed to enhance the findings of this study.

Our study indicated that Pre-K attendance had a positive effect on childhood asthma at least among financially disadvantaged populations. Previous surveys on childhood asthma in the United States have revealed a gradient prevalence of asthma by financial conditions of families: 6.8% in the most affluent group and 11% in the most challenged group [44]. As financial disadvantage is associated with a higher rate of smoking habits [30], tobacco-free air conditions at Pre-K schools may be beneficial in preventing childhood asthma in the long run, especially among children from financially disadvantaged families. Further investigation of whether the health impact of Pre-K differs according to the socioeconomic status of families is needed.

This study had several methodological strengths. The propensity score weighting used in our DID model is one of the most rigorous ways to investigate a causal inference in observational research [39]. As discussed in the Materials and Methods section, our dataset had a good fit to the DID model and displayed an adequate balance between the intervention group and the non-treatment groups. We conducted our study fully aware that some conditions that cause regression to the mean bias can threaten DID analysis with propensity score modeling [41]. Although the two groups in our dataset are different in outcome level, they do not directly violate the validity of the DID analysis. In addition, we selected time-invariant covariates for propensity score modeling and confirmed that none of the covariates had significant associations with the outcomes. Thus, our modeling may not overestimate the results (see Appendix E: DID analysis without using propensity score modeling for a robustness check).

Our longitudinal panel analysis, using a fixed-effect model, accounted for unobservable heterogeneity across participants. One of the most powerful heterogeneities that can affect childhood asthma is hereditary and inherited from their biological parents [45]. Although we did not have information on parental history of asthma, our fixed-effect model suggests that this lack of information does not undermine the integrity of our findings.

### 4.2. Limitations

This study has several limitations. We did not know whether children and their caregivers used other services (e.g., Head Start) prior to Pre-K in the current study. If a child had attended Early Head Start, Head Start, or private childcare centers before Pre-K, these settings might have provided other opportunities for surveillance of asthma symptoms and other illnesses. Given that such educational opportunities prior to Pre-K were equally accessible to the treated and non-treated children in this study, children’s participation in these programs might not alter our findings. However, further investigation of the Pre-K participants’ educational/childcare experience before Pre-K is needed to generalize our findings. Regional diversity such as accessibility to childcare settings, state (city) budget restrictions, and eligibility criteria for Pre-K participation in the state (city) should also be carefully considered.

The different time points of the participants’ enrollment in and disenrollment from Medicaid throughout the study period is another limitation that may threaten our propensity score modeling as well as our estimation of Pre-K’s effects. For example, different enrollment rates in Medicaid between the treatment group and the non-treatment group before the Pre-K period may have impeded our ability to capture children’s pre-existing health conditions. Our additional analysis that restricted participants’ continuous enrollment in Medicaid during the study period showed results that were highly consistent with our main findings (see Appendix G). We consider our findings to be robust to some extent because we tested Pre-K’s effects in different time windows (i.e., before, during, and after intervention periods).

Potential selection bias caused by unobservable factors that could influence the selection process to attend the Pre-K program could be another limitation of this study. As we discussed above, our panel analysis with a fixed-effect model would counter the confounding effect of unobservable and time-invariant factors such as hereditary characteristics and regional air quality on the outcome. Although we believe these factors are unlikely to affect the selection process of Pre-K participation, panel analysis with propensity matching could not eliminate selection bias caused by unobserved factors. Future studies that enable other approaches to rigorously address this potential selection bias are needed to confirm our results.

This study also did not examine exposure to environmental asthma triggers such as allergic substances and indoor air quality [46]. To account for the effects of Pre-K programs on asthma, it is also important to evaluate the air quality in pre-primary schools. Future research on Pre-K programs could quantitatively assess indoor air quality parameters.

In addition, the generalizability of our findings is limited. As every public-funded pre-K program is unique in terms of program design, scale, population served, and demographic characteristics of enrolled children, the application of our findings to other states and cities should be carefully considered. Although we identified some plausible pathways that explain the association between Pre-K and childhood health, the mechanism by which Pre-K affected the diagnosis and treatment of asthma in children could not be addressed in this study. Our stratified analysis of children’s pre-existing asthma-related conditions sheds light on the possibility that Pre-K attendance may have different impacts on children according to their history of disease before the intervention period. The detailed mechanism of how Pre-K affects children’s health outcomes, which may subsequently reveal Pre-K’s differential health impact by demographic or constitutional features of the children, should be investigated in further studies, particularly if such studies are to inform future policy decisions.

### 4.3. Policy Implication

This study has significant implications for public health policy and planning. A state-funded Pre-K program in NC affected the asthma-related health service use of the children who enrolled in it by enabling school professionals to detect early symptoms and promote adequate disease management for the children they supervised. Early detection of pediatric asthma could decrease medical expenditures in the long term. Universal pre-primary education offers as a form of investment in child health, which is an important means of ensuring children’s future human capital. 

## 5. Conclusions

The findings of this study support the hypothesis that Pre-K can increase the use of health services related to asthma during the program and decrease the service use after the enrolled children leave the program. The magnitude of the decrease after the program period was greater than the increase during it among the Pre-K participants. These findings suggest that attending Pre-K may have a detection effect, in which school professionals identify symptoms in children and encourage caregivers to seek health services for them. Pre-K may also have a preventive effect on childhood asthma in the long run due to the early detection and subsequent early initiation of medical care for the symptoms. 

## Figures and Tables

**Table 1 ijerph-18-10461-t001:** Characteristics of the participants.

Characteristics		Treatment Group (*n* = 279)		Non-Treatment Group (*n* = 333)		*p*-Value ^1^
		*n*/mean	%/SD	*n*/mean	%/SD	
Gender	Male	144	52%	142	43%	0.027 *
Race/ethnicity	Non-white	152	54%	161	48%	0.131
Having chronic disease	Yes	39	14%	50	15%	0.717
Child speaks English at home	No	62	22%	71	21%	0.788
Child’s first language is English	No	58	21%	62	19%	0.501
Parent speaks English at home	Yes	62	22%	69	21%	0.652
Military family	Yes	32	11%	22	7%	0.035 **
Parental education	Less than High School	65	23%	58	17%	0.177
	High School diploma	161	58%	202	61%	
	Any college	53	19%	73	22%	
Family size		4.33	1.40	4.43	1.32	0.369
Number of adults		1.89	0.74	1.88	0.75	0.835
Number of children		2.52	1.12	2.68	1.25	0.099 *
Family income ($)		7244.8	2600.4	7122.6	2602.0	0.563
Neonatal problems		5	2%	10	3%	0.335
Behavioral problems		87	31%	97	29%	0.581
Metabolic diseases		21	8%	9	3%	0.006 **
Asthma-related conditions		73	26%	72	22%	0.188
Abnormal weight		9	3%	6	2%	0.257
History of routine health checkups (times)		4.20	1.92	3.64	2.28	0.001 **
Population density of the county (/square mile)		270.6	317.9	366.0	379.2	0.001 **

^1^ Chi-square test for categorical variables and *t*-test for numeric variables; * *p* < 0.1; ** *p* < 0.05.

**Table 2 ijerph-18-10461-t002:** Proportion of asthma diagnoses in a person-month unit in three periods.

Periods and Asthma Categories	Observation (*n* = 51,408)	Treatment Group (*n* = 279)		Non-Treatment Group (*n* = 333)		*p*-Value ^1^
		*n*	%	*n*	%	
All asthma						
Before Pre-K period	26,928	249	2.03	196	1.34	<0.001 ***
During Pre-K period	7344	119	3.55	67	1.68	<0.001 ***
After Pre-K period	17,136	143	1.83	202	2.17	0.119
Severe asthma						
Before Pre-K period	26,928	75	0.61	53	0.36	0.003 **
During Pre-K period	7344	34	1.02	16	0.40	0.001 **
After Pre-K period	17,136	68	0.87	69	0.74	0.340
Mild asthma						
Before Pre-K period	26,928	197	1.60	169	1.15	0.001 **
During Pre-K period	7344	100	2.99	56	1.40	<0.001 ***
After Pre-K period	17,136	84	1.08	147	1.58	0.005 **

^1^ Chi-square test for categorical variables and *t*-test for numeric variables; ** *p* < 0.05; *** *p* < 0.001

**Table 3 ijerph-18-10461-t003:** Predicted coefficient of the interaction term of treated groups and period; pre vs. during and during vs. after.

	Coefficient	*p*-Value	95% Confidence Interval	
**All asthma**				
Age of service use (month)	−0.0001	0.404	−0.0002	0.0001
Before period	−0.0060	0.103	−0.0133	0.0012
After period	0.0047	0.292	−0.0040	0.0134
Treated group × Before period	−0.0145	0.058 *	−0.0295	0.0005
Treated group × After period	−0.0216	0.002 **	−0.0353	−0.0080

**Severe asthma**				
Age of service use (month)	−0.0001	0.076 *	−0.0001	0.0000
Before period	−0.0018	0.259	−0.0048	0.0013
After period	0.0056	0.006 **	0.0016	0.0096
Treated group × Before period	−0.0061	0.03 **	−0.0116	−0.0006
Treated group × After period	−0.0064	0.025 **	−0.0120	−0.0008

**Mild asthma**				
Age of service use (month)	0.0000	0.753	0.000	0.000
Before period	−0.0044	0.183	−0.011	0.002
After period	0.0040	0.993	−0.008	0.008
Treated group × Before period	−0.0120	0.099 *	−0.026	0.002
Treated group × After period	−0.0185	0.007 **	−0.032	−0.005

* *p* < 0.1; ** *p* < 0.05, ‘×’ indicates interaction

## Data Availability

Data use is strictly limited to those with official approval from the NC Division of Health Benefits. Data for this study are de-identified data provided by the NC Division of Health Benefits and are not publicly available. Inquiries for data content should be directed to the authors and may be available subject to appropriate approval from the NC Division of Health Benefits.

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
