# Peer review of "A Quasi-Experimental Study of the Effects of Pre-Kindergarten Education on Pediatric Asthma"

_ijerph, 2021, doi:10.3390/ijerph181910461_

Round 1
Reviewer 1 Report
The study is well designed and presented. Results do not raise any doubts. The purpose of this study is to investigate the causal effects of a Pre-K program on pediatric asthma. Children’s individual data from existing research conducted in North Carolina was linked with state Medicaid 16 claims data from 2011–2017. There were 279 children enrolled in Pre-K and 333 unenrolled children. Authors selected three time periods (i.e., before, during, and after Pre-K) to investigate the effect of Pre-K enrolment on asthma diagnosis and treatment.
A wide range of parental characteristics were derived from the original Pre-K evaluation study. The study analysed the period of 84 months of data for each of the 612 children included in the study. Pre -K increased rates of asthma diagnosis and decreased rates of asthma-related health service use among children who attended Pre-K. The increase in asthma diagnosis during Pre-K period may be attributable to the detection effect of Pre-K. Some plausible pathways that explain the association between Pre-K and child health.
Study is vet valuable,as it provides some evidence on protestation of Pe-K for long term asthma.
Author Response
I appreciate the comment. We hope this paper provides researchers/policymakers with valuable information regarding the effects of Pre-K programs on children’s health.
Reviewer 2 Report
A brief summary:
This manuscript used Medicaid claims data from 2011-2017, and estimated the propensity scores of Pre-K enrollment. The authors measured the impact of Pre-K enrollment on children's asthma diagnosis. Results indicated that Pre-K enrollment can help increase detection of asthma symptoms.
Broad comments:
Diagnosing asthma in children might be difficult in a home because the symptoms of asthma can be confused with those of other respiratory diseases. During pre-primary education, the detection of diseases by teachers having asthma knowledge seems to be important.
Previous studies have shown that patients with lower socioeconomic status have higher asthma morbidity, and Medicaid-insured children have higher emergency department visits than commercially insured children [1].
In line with the previous studies, we can consider the hypothesis that the symptoms of asthma of Medicaid-insured children can be found through Pre-K education, and children's health status may be improved after physician visits. Thus, I think this study is well-designed. Propensity score matching method with DID analysis is a good empirical strategy.
On the other hand, it is well known that the educational intervention to children having asthma contributes to improve children's health outcome.
Smith et al. [2] used data of children age 4 to 17 with at least one month of Medicaid eligibility among 40 states drawn from the 2012 Medicaid Analytic eXtract files, they identified 17 states with lower diagnosed asthma prevalence in the Medicaid population as compared to the Centers for Disease Control and Prevention estimates.
As Medicaid funded children need no medication co-payments, the educational intervention by teachers having asthma knowledge may play the most important role to detect children's diseases. Indeed, school-based self-management interventions for asthma among primary education children can improve asthma outcomes [3].
I have two major comments regarding areas of weakness. One is the role of teachers in Pre-K, another is the selection effects on unobserved variables.
(1) Readers cannot understand whether the educational intervention contributed to improve children's health outcome.
Is there any evidence that teachers in Pre-K have known certain predictors for asthma such as a family history of allergies or asthma, prenatal and postnatal exposure to tobacco smoke, living in an area with high air pollution, or obesity?
How often did teachers in Pre-K who have known certain predictors for asthma detect children's diseases?
Please add the relevant studies into the manuscript, if any.
(2) Because an estimated fixed-effect model included unobservable heterogeneity such as hereditary factors, the authors must explain the correlation between Treated (children enrolled in Pre-K) and unobservable individual’s effect.
Although the assumption of fixed-effect model allows the correlations between individual’s effect and independent variables, the possible bias of Treated is not small when there are significant relationships between both variables.
This manuscript ignores the selection effect driven by unobserved respondent characteristics such as living in an area with high air pollution. I think the selection effects on unobserved variables should be considered.
When parents living in an area with high air pollution have a history of allergies or asthma, the low likelihood is given to the coefficient of Treated.
The selection factor which depends on unobserved characteristics may become more prominent [4].
Propensity scores matching is only successful when there is no selection on unobserved variables. Following the procedure of [4], stratification matching may be helpful for your study.
Please re-consider the selection effect driven by unobserved respondent characteristics.
References
[1] Andrews, AL et al. (2019) A Longitudinal Examination of the Asthma Medication Ratio in Children with Medicaid. Journal of Asthma. DOI: 10.1080/02770903.2019.1640727.
[2] Smith, A BS et al. (2019) Asthma Prevalence Among Medicaid-Enrolled Children J Allergy Clin Immunol Pract. 7(4):1207-1213.e4. DOI: 10.1016/j.jaip.2018.10.008.
[3] Ramdzan, SN et al. (2021) School-based self-management interventions for asthma among primary school children: a systematic review. npj Primary Care Respiratory Medicine 31:18.
DOI: 10.1038/s41533-021-00230-2
[4] Capacci, S et al. (2018) Estimation of unobservable selection effects in on-line surveys through propensity score matching: An application to public acceptance of healthy eating policies. PLOS ONE 13(4): e0196020. DOI: 10.1371/journal.pone.0196020
Author Response
Point 1: Readers cannot understand whether the educational intervention contributed to improve children's health outcome. Is there any evidence that teachers in Pre-K have known certain predictors for asthma such as a family history of allergies or asthma, prenatal and postnatal exposure to tobacco smoke, living in an area with high air pollution, or obesity? How often did teachers in Pre-K who have known certain predictors for asthma detect children's diseases? Please add the relevant studies into the manuscript, if any.
Response 1:
We appreciate the very helpful comment regarding the capability of Pre-K teachers to detect children’s health problems. Under the rules of NC Pre-K and child care licensing [1], for teachers, there are training requirements related to health and safety. The relevant topics include (1) Prevention and control of infectious diseases, including immunization; (2) Administration of medication, with standards for parental consent; (3) Prevention of and response to emergencies due to food and allergic reactions. The training on all topics needs to be completed every 5 years. Teachers are responsible for safe practices with regard to the administration of medication and emergency health needs in daily activities.
We should also explain the role of licensed health professionals in Pre-K. NC Pre-K programs are required to have health screenings (conducted by licensed health care professionals) on file within 30 days of a child’s enrollment and follow any required recommendations. This health screening includes a child’s health condition within the past 12 months and emergency information. Both of these could include information related to symptoms of asthma. Teachers can refer to a child’s file when needed.
In sum, teachers are to some extent trained to find a child’s present health problems in daily activities. In addition, health screenings conducted by health professionals may also play an important role to detect children’s health problems including asthma. Following the reviewer’s advice, we now consider the role of health professionals’ health assessments, and subsequent referrals should be included in the manuscript. We added an explanation regarding this to “Description of the Pre-K Program” as follows (underbar indicates words/sentences we edited/added. Lines are revised ones):
<Section 2.1. lines: 160-162>
Specifically, NC Pre-K programs are required to have health screenings conducted by licensed healthcare professionals within 30 days of a child’s enrollment and follow any required recommendations.
Also, we revised the manuscript as follows (underbar indicates words/sentences we edited/added. Lines are revised ones):
<Section 1.2. lines: 107-109>
First, attending pre-primary education increases the detection of diseases by teachers and licensed health professionals.
<Section 4.1: lines 377-379 >
Teachers’ observations and mandated health check-ups by licensed health professionals might be crucial occasions for detecting pertinent symptoms such as coughing…
< Section 4.1: lines 385-387 >
Pre-K teachers’ and health professionals’ abilities to detect diseases in children and connect their caregivers to relevant health services is another beneficial side effect of Pre-K programs.…
< Section 4.1 lines:396-399 >
Children who enrolled in the Pre-K program received a detection effect 1 year earlier than their non-enrolled counterparts because (in principle) all children should be equally observed by school professionals when they begin kindergarten. If Pre-K enrollment allows the professionals’ detection of asthma symptoms to begin 1 year earlier…
< Section 4.1 lines: 410-413 >
Due to teachers’ and health professionals’ advice, caregivers of children enrolled in Pre-K possibly started routine monitoring of asthma symptoms earlier and taking advantage of the results from required health assessments as well as any follow-up evaluations recommended by the Pre-K program…
< Section 4.3. lines: 509-512>
A state-funded Pre-K program in NC affected the asthma-related health service use of the children who enrolled in it by enabling school professionals to detect early symptoms and promote adequate disease management for the children they supervised.
<Conclusion lines: 520-523>
These findings suggest that attending Pre-K may have a detection effect, in which school professionals identify symptoms in children and encourage caregivers to seek health services for them.
[1] NC Department of Health and Human Services, NC Pre-Kindergarten (NC Pre-K) Program Requirements and Guidance. 2014. Available online: http://www.guilfordchildren.org/wp-content/uploads/2014/06/NC-Pre-K-Program-Requirements-2014-2015FINAL.pdf
Point 2: Because an estimated fixed-effect model included unobservable heterogeneity such as hereditary factors, the authors must explain the correlation between Treated (children enrolled in Pre-K) and unobservable individual’s effect. Although the assumption of a fixed-effect model allows the correlations between individual’s effect and independent variables, the possible bias of Treated is not small when there are significant relationships between both variables. This manuscript ignores the selection effect driven by unobserved respondent characteristics such as living in an area with high air pollution. I think the selection effects on unobserved variables should be considered. When parents living in an area with high air pollution have a history of allergies or asthma, the low likelihood is given to the coefficient of Treated. The selection factor which depends on unobserved characteristics may become more prominent [4]. Propensity scores matching is only successful when there is no selection on unobserved variables. Following the procedure of [4], stratification matching may be helpful for your study.
Please re-consider the selection effect driven by unobserved respondent characteristics.
Response 2:
We appreciate the comment on potential selection biases. We should admit we have only limited information for propensity score calculation, and there should have been several unobservable factors that may generate a selection effect. In the study, we addressed this problem by applying panel analysis with a fixed-effect model, which eliminates the effect of unobservable and time-invariant factors such as parental hereditary factors on the outcome [2]. We consider the regional air pollution level as one of such unobserved factors that should be well dealt with in our analytical model. Although we notice that regional air pollution level is essentially time-variant, we assumed there was not a significant shot-term change during the study period (2011-2017) in the state. Thus, we consider we covered the lack of information by our analytical model.
However, we should admit that if these unobservable yet time-invariant factors were influential on the selection process to Pre-K inclusion, it may cause a remaining selection bias. Although we do not think of good reasons that hereditary characteristics, regional air quality, and other time-invariant factors such as regional accessibility of asthma clinic and regional socio-economic/ethnic characteristics, can be influential on the selection of an individual child for Pre-K inclusion independent of existing inclusion criteria.
To overcome this selection bias, other approached such as a regression discontinuity or randomized control trial should be helpful. Unfortunately, our dataset has only the same-year-age children. We do think further study dataset including different (and continuous)-year-age children, which enables such as RD approach, is needed.
Following the reviewer’s advice and the above thoughts, we note this potential bias as one of our study limitations. We revised the manuscript as follows (Lines are revised ones):
<Section 4.2. lines: 481-489>
Potential selection bias caused by unobservable factors that could influence the selection process to attend the Pre-K program could be another limitation of this study. As we discussed above, our panel analysis with a fixed-effect model would counter the confounding effect of unobservable and time-invariant factors such as hereditary characteristics and regional air quality on the outcome. Although we believe these factors are unlikely to affect the selection process of Pre-K participation, panel analysis with propensity matching could not eliminate selection bias caused by unobserved factors. Future studies that enable other approaches to rigorously address this potential selection bias are needed to confirm our results.
[2] Wooldridge, J. M., Introductory econometrics: a modern approach. 7th ed ed.; Cengage: 2020; p xxii, 826, pp. 413-435
[3] Weiland, C.; Yoshikawa, H., Impacts of a prekindergarten program on children's mathematics, language, literacy, executive function, and emotional skills. Child Dev 2013, 84 (6), 2112-30.
Reviewer 3 Report
This paper addresses an important, under-researched topic pertaining to the impact of asthma-related pre-K education on asthma healthcare utilization in the US (based on data and experience in the state of North Carolina). The study’s significance, methodology, and results are well-described, including the detailed analytic techniques associated with the Difference-in-Difference approach to panel data regression analysis. The study’s conclusions are appropriately inferred from results; and limitations and implications are well-discussed.
I only have some minor revisions to recommend to authors to improve the flow and organization of this paper.
- The Introduction is a large chunk of text covering a variety of areas, including study background, literature review, gaps in literature (limitations of earlier studies), study significance and rationale, culminating in the study purpose and research questions. This section would greatly benefit from sub-headings to guide the reader in understanding the study background and significance. Also, it would help to have a separate subheading for study purpose and to indent and present the research questions in bulleted format to highlight the scope and aims of the study.
- I would recommend similar revisions to the Discussion section to include separate subheadings for “summary of findings,” “study limitations,” implications for practice & policy,” and implications for future research.
- A data availability statement also needs to be included.
Thank you for the opportunity to review this interesting paper!
Author Response
Point 1: The Introduction is a large chunk of text covering a variety of areas, including study background, literature review, gaps in literature (limitations of earlier studies), study significance and rationale, culminating in the study purpose and research questions. This section would greatly benefit from sub-headings to guide the reader in understanding the study background and significance. Also, it would help to have a separate subheading for study purpose and to indent and present the research questions in bulleted format to highlight the scope and aims of the study.
Response 1:
Following the suggestion, we separated the Introduction into 3 sub-headings (1.1. Background, 1.2. Underlying Mechanism, and 1.3 Study Purpose). For the aim of this study, we already put (a) and (b) to distinguish two different research questions, thank you.
Point 2: I would recommend similar revisions to the Discussion section to include separate subheadings for “summary of findings,” “study limitations,” implications for practice & policy,” and implications for future research.
Response 2:
Following the suggestion, we separated the Discussion into 3 sub-headings (4.1. Study Findings, 4.2. Limitations, and 4.3 Policy Implications).
Point 3: A data availability statement also needs to be included.
Response 3:
We added the data availability statement as follows (see section 6. Patent):
“Data use is strictly limited to those with official approval from the NC Division of Health Benefits. Data for this study are de-identified data provided by the NC Division of Health Benefits and are not publicly available. Inquiries for data content should be directed to the authors and may be available subject to appropriate approval from the NC Division of Health Benefits.”
We appreciate your helpful comments.
Reviewer 4 Report
Thank you for opportunity to review this paper. Revisions are warranted.
Please refer to the attached PDF file with comments ("sticky notes") to correspond to highlighted sections and lines numbered. [Please note some highlighted sections were only out of my own interest, and I liked/agreed with what was written there unless comment provided.]

Round 2
Reviewer 2 Report
The role of licensed health professionals in Pre-K is explained. I have no further comments.
Author Response

(The authors gave the same response as above.)

Reviewer 4 Report
Thank you for revising your manuscript.
It seems you re-wrote the Abstract (I have re-read it and there are differences); however, overall, it has improved and addresses prior concerns. There does seem to be one word missing in line 22: "...enrolled children had a lower risk of receiving...." Insert word "risk" thank you.
Otherwise, the authors did a good job in responding to my 1st review comments, and adding text to different parts of Introduction, Methods and Discussion section as well as the Conclusions paragraph.
Author Response
We sincerely appreciate your careful reading. We added "risk" in line 22. Again, thank you very much for your advice.